# Prevalence of and risk factors for chronic kidney disease of unknown aetiology in India: secondary data analysis of three population-based cross-sectional studies

Cristina O'Callaghan-Gordo,[1,2,3,4] Roopa Shivashankar,[5,6] Shuchi Anand,[7] Shreeparna Ghosh,[5] Jason Glaser,[4,8] Ruby Gupta,[5] Kristina Jakobsson,[9,10] Dimple Kondal,[5,6] Anand Krishnan,[11] Sailesh Mohan,[5] Viswanathan Mohan,[12,13] Dorothea Nitsch,[14] Praveen P A,[6,15] Nikhil Tandon,[15] K M Venkat Narayan,[16] Neil Pearce,[4,17] Ben Caplin,[18] Dorairaj Prabhakaran[5,6]

CO'C-G and RS are joint first authors.
BC and DP are joint last authors.

For numbered affiliations see end of article.

**Correspondence to**
Dr Cristina O'Callaghan-Gordo; cristina.ocallaghan@isglobal.org

## ABSTRACT

**Objectives** To assess whether chronic kidney disease of unknown aetiology (CKDu) is present in India and to identify risk factors for it using population-based data and standardised methods.

**Design** Secondary data analysis of three population-based cross-sectional studies conducted between 2010 and 2014.

**Setting** Urban and rural areas of Northern India (states of Delhi and Haryana) and Southern India (states of Tamil Nadu and Andhra Pradesh).

**Participants** 12 500 individuals without diabetes, hypertension or heavy proteinuria.

**Outcome measures** Mean estimated glomerular filtration rate (eGFR) and prevalence of eGFR below 60 mL/min per 1.73 m$^2$ (eGFR <60) in individuals without diabetes, hypertension or heavy proteinuria (proxy definition of CKDu).

**Results** The mean eGFR was 105.0±17.8 mL/min per 1.73 m$^2$. The prevalence of eGFR <60 was 1.6% (95% CI=1.4 to 1.7), but this figure varied markedly between areas, being highest in rural areas of Southern Indian (4.8% (3.8 to 5.8)). In Northern India, older age was the only risk factor associated with lower mean eGFR and eGFR <60 (regression coefficient (95% CI)=−0.94 (0.97 to 0.91); OR (95% CI)=1.10 (1.08 to 1.11)). In Southern India, risk factors for lower mean eGFR and eGFR <60, respectively, were residence in a rural area (−7.78 (−8.69 to −6.86); 4.95 (2.61 to 9.39)), older age (−0.90 (−0.93 to −0.86); 1.06 (1.04 to 1.08)) and less education (−0.94 (-1.32 to −0.56); 0.67 (0.50 to 0.90) for each 5 years at school).

**Conclusions** CKDu is present in India and is not confined to Central America and Sri Lanka. Identified risk factors are consistent with risk factors previously reported for CKDu in Central America and Sri Lanka.

## Strengths and limitations of this study

► The use of a random selection of population-based participants allows the estimation of chronic kidney disease of unknown aetiology (CKDu) prevalence in the general population.
► A large sample size including participants from different areas of India (urban and rural, and Northern and Southern India) increases the representativeness of the results.
► The use of standardised definitions of CKDu facilitates international comparisons of CKDu prevalence and risk factors.
► The prevalence of estimated glomerular filtration rate <60 observed in this study is likely to be underestimated; however, this is unlikely to have biased the internal comparisons conducted in this study.

## INTRODUCTION

High prevalence of chronic kidney disease of unknown aetiology (CKDu) has mainly been reported in the last decades among the working age populations of agricultural communities of tropical/subtropical regions, specifically in Central America and Sri Lanka.[1–3] In Nicaragua and El Salvador, the estimated prevalence of estimated glomerular filtration rate (eGFR; the clinical measure of kidney function) below 60 mL/min per 1.73 m$^2$ (eGFR <60), in the absence of diabetes and hypertension, was 10%–20%.[4–6] It has been suggested that CKDu may also be highly prevalent in other low-income and middle-income countries (LMICs), including India.[7–11] However, it is not clear in which other regions of the world CKDu occurs, whether the underlying aetiology is the same in different regions and what the risk factors are. Currently, there is no consensus, but factors such as heat stress,

strenuous work, climatic conditions, agrochemical use, heavy metal exposure and infections have been suggested as risk factors.[1 12–15]

Data on CKDu from India are scarce. The recent report of verbal autopsy data from India suggests CKD of all causes is a growing problem. However, it does not provide accurate population-based data on CKDu.[16 17] Existing reports indicate that CKDu may be common but it is difficult to be definite about this because of the absence of population-based studies using standardised and comparable methods. Data from the Indian CKD Registry, a hospital based registry of incident cases of CKD between 2006 and 2010, found that CKDu was the second most common form of CKD after diabetic nephropathy.[10] However, this is restricted to referred cases and therefore may not be representative of the general population. There are also sporadic reports of high numbers of CKDu cases among agricultural communities of the South Eastern Indian states of Andhra Pradesh and Odisha (reviewed by Chatterjee[18] and Ganguli[19]). However, population-based data have not been reported for India.

We conducted a secondary analysis of representative sample surveys conducted in India between 2010 and 2014. Given the absence of a clear case definition for CKDu it is necessary to make a presumptive diagnosis based on measures/estimates of GFR in the absence of known risk factors for kidney disease. The overall aim of the current study was to use a methodology which is comparable with previous studies elsewhere in the world (particularly in Central America) to assess the extent to which reduced kidney function is a problem in India, and which areas and subpopulations are most affected. We therefore: (1) assessed the distribution eGFR and prevalence of eGFR below $60\,mL/min$ per $1.73\,m^2$ (eGFR <60) in Indian populations restricted to those without known risk factors for CKD, i.e. diabetes, hypertension or heavy proteinuria; (2) compared these outcomes in North and South India and in urban and rural populations; and (3) identified the risk factors associated with these outcomes.

## METHODS
### Study population
We used cross-sectional data from three population-based studies conducted in India: the 'Centre for Cardiometabolic Risk Reduction in South Asia' cohort study (CARRS study),[20] the 'Implementing a comprehensive diabetes prevention and management program' study (UDAY study)[21] and the 'Prevalence of Coronary Heart Disease repeat survey' study funded by the Indian Council of Medical Research (ICMR-CHD study).[22] Details on study design and selection of participants from the CARRS, UDAY and ICMR-CHD studies have been previously described[20–22] and are summarised in table 1.

For the current analyses, we excluded participants with missing information on serum creatinine, as this variable was necessary to estimate eGFR. As the focus of our study was CKDu, we excluded participants with known

risk factors for CKD (ie, diabetes and hypertension) or evidence of primary glomerular disease (as assessed by heavy proteinuria) or with missing information for these risk factors. We also excluded participants with missing information on basic covariables (education) for all the analyses conducted. A study flowchart is presented. We classified participants as having: diabetes, if plasma fasting glucose was ≥126 mg/dL or glycated haemoglobin A1c (HbA1c) was ≥6.5% or self-reported diabetes; hypertension, if systolic blood pressure was ≥140 mm Hg or diastolic blood pressure was ≥90 mm Hg or self-reported hypertension; and heavy proteinuria, if the albumin:creatinine ratio (ACR) in urine was ≥300 mg/g. We used the CKD-EPI equation to estimate GFR.[23]

### Data collection and laboratory analyses
Data collection was conducted between October 2010 and December 2014. All three studies used a standardised questionnaire to collect data on age, sex, completed years of education (0, ≤5, >5–≤10, >10), alcohol intake (ever, never) and dietary habits (vegetarian yes, no). Weight, height and body composition were measured using stadiometers (SECA 214 in the three studies) and electronic bioimpedance measuring instruments (Tanita BC 418 in CARRS and ICMR-CHD studies, and Tanita BC 601 in UDAY study). Body mass index (BMI, $kg/m^2$) was calculated and categorised (≤18.5: underweight; >18.5–≤25: normal weight; >25–≤30: overweight; >30: obese) and fat free mass was derived from bioelectric impedance analysis. In CARRS and ICMR-CHD studies, fat-free mass (kg) was directly measured as previously described,[24] whereas in UDAY study, fat free mass was estimated from the percentage of total body fat. To estimate fat-free mass from the percentage of body fat, we calculated the amount of total body fat by multiplying the percentage of body fat by the weight of the participant, and from that value we estimated the amount of fat-free mass by subtracting the weight of total body fat from the total weight of the participant. Blood pressure was measured using electronic sphygmomanometers (OMRON (HEM-7080) in CARRS and ICMR-CHD studies, and OMRON (HEM 7200) in UDAY study), as previously reported.[20 25] Stadiometers, electronic bioimpedance measuring instruments, and electronic sphygmomanometers were calibrated before each study, and no re-calibration was needed during the duration of different studies. A fasting venous blood sample was used to measure glucose levels, HbA1c and serum creatinine levels and urine sample to measure albuminuria and creatinuria.[20] Glucose levels were measured using hexokinase/kinetic methods, HbA1c using high-performance liquid chromatography, serum creatinine using the rate-blanked and compensated kinetic Jaffe method, traceable to isotope dilution mass spectrometry and albuminuria using immune turbidmetric method.[20] Samples from UDAY, ICMR-CHD and samples from CARRS from Delhi were analysed at Public Health Foundation of India (PHFI) laboratory and samples from CARRS from Chennai were analysed

**Table 1** Design and methods of the three studies included in the current analysis

| | CARRS | | UDAY | | ICMR-CHD | |
|---|---|---|---|---|---|---|
| Latitude (north/south) | North | South | North | South | North | North |
| Residence (urban/rural) | Urban | Urban | Urban | Rural | Urban | Rural |
| District (and state) | Delhi (National Capital Territory of Delhi) | Chennai (state of Tamil Nadu) | Sonipat (state of Haryana) | Vishakhapatnam (state of Andhra Pradesh) | Delhi (National Capital Territory of Delhi) | Faridabad (state of Haryana) |
| Household sampling | Multistage cluster random (wards—census enumeration blocks—households) | | Multistage cluster random (census enumeration blocks (urban) or villages (rural)—households) | | Multistage cluster random (wards—census enumeration blocks—households) | Simple cluster random (based on health and demographic surveillance system) |
| Individual sampling | One man and one woman from each household (selected by Kish method[43])* | | One man and one woman from each household (selected by Kish method[43])* | | All adults | |
| Age groups included | ≥20 | | ≥30 | | ≥30 | |
| Exclusion criteria | Pregnant, bedridden and participants who were unable to comprehend the questionnaires due cognitive deficiencies were excluded | | | | | |
| Study period | October 2010–November 2011 | | July 2014–December 2014 | | August 2010–January 2012 | |
| Laboratory† | PHFI | MDRF | PHFI | | PHFI | |

* In households where only eligible men or only eligible women were present, we selected just one adult.
† Study laboratories participated in RIQAS for clinical chemistry and HbA1c during the entire study periods.
CARRS, Centre for Cardiometabolic Risk Reduction in South Asia; HbA1c, glycated haemoglobin A1c; ICMR-CHD, Indian Council of Medical Research-Coronary Heart Disease; MDRF, Madras Diabetes Research Foundation; PHFI, Public Health Foundation of India; RIQAS, Randox International Quality Assurance Scheme.

at Madras Diabetes Research Foundation (MDRF) laboratory. Both PHFI and MDRF laboratories used the same methodologies and protocols to analyse the samples and participated in Randox International Quality Assurance Scheme for clinical chemistry and HbA1c during the entire study periods. Data from the three studies were homogenised and merged in a single data set.

### Statistical analyses

We reported mean eGFR and prevalence of eGFR <60 according to different characteristics of the study populations. UDAY and CARRS studies did not involve fully random population samples (since sampling was based on households, with one participant per household) and the proportions of study participants with particular outcomes (eg, eGFR <60), will not be exactly the same (but very similar) to what would have been obtained with genuine random population samples; thus in this paper we refer to the prevalence in the study participants, not overall population prevalence estimates. We used linear regression models to estimate the associations between potential risk factors and eGFR and logistic regression models to estimate the associations between potential risk factors and eGFR <60. We also repeated the analyses separately for males and females. Variables associated with eGFR in the basic analyses (adjusted for age and sex) were considered for the multiple regression analysis. In the final multiple regression model, we included all variables that were of a priori interest and/or had shown independent associations with eGFR. We then checked for multicollinearity for each variable in the multiple regression analyses in comparison with the basic analyses.[26] Six per cent of participants had missing values for basic co-variables (ie, education) and were excluded from the analysis; 5% and 9% of participants had missing values for BMI and for fat-free mass, respectively. These participants were included in the main analysis, but we excluded them to compare models non-adjusted and adjusted for these variables. We calculated prevalence ratios of eGFR <60 for rural versus urban areas in different age groups. Urban areas were defined as 'all places with a municipality, corporation, cantonment board or notified town area committee, etc., and all other places which satisfied the following criteria: a minimum population of 5,000; at least 75 per cent of the male main working population engaged in non-agricultural pursuits; and a density of population of at least 400 persons per km$^2$, according to the 2011 Census of India definition.[27] Finally, we estimated potential interactions between urban (versus rural) residence and latitude (Northern India (ie, states of Delhi and Haryana) versus Southern India (states of Tamil Nadu and Andhra Pradesh). Classification of latitude was done in concordance with the classification of major geographical areas on India defined by the ICMR.[28] We conducted all analyses using Stata V.14 (StataCorp).

### Patient and public involvement

Patients were not involved in the design of this analysis.

### Results

#### Characteristics of study participants

A total of 12 500 people were eligible for the current analyses (figure 1). Table 2 summarises the sociodemographic and anthropometric characteristics of the 12 500 study participants included in this analysis (the same information including participants with known risk factors for CKD (n=24 774) in online supplementary material table S1). The mean (standard deviation (±SD)) age of participants was 41.5±11.6 years. 88% (4805/5434) of the male population was formally employed; 76% (5346/7066) of women worked on house duties (ie, housewives). The mean BMI was 24±5.0 kg/m$^2$ and mean fat free mass was 42±15 kg/m$^2$. The mean fasting plasma glucose was 91.9±12.3 mg/dL and the mean HbA1c was 5.5%±0.4%. The mean systolic and diastolic blood pressures were 114±12 mm Hg and 74±9 mm Hg, respectively. The median (IQR) ACR was 2.4 (4.3) mg/g (after exclusion of those with ACR >300 mg/g, n=1208).

#### Mean eGFR and prevalence of eGFR <60

The mean eGFR was 105.0±17.8 mL/min per 1.73 m$^2$. The mean eGFR was lower at increasing ages, in males, in inhabitants from rural areas and in those from Northern India, in participants with no formal education, and in participants who reported tobacco consumption, alcohol intake and being vegetarian (table 2). We observed differences in mean eGFR depending on the area, being 104.5±17.6 in urban areas of Northern India, 100.3±16.2 in rural areas of Northern India, 110.9±15.7 in urban areas of Southern India and 97.4±19.8 in the rural area of Southern India.

The prevalence of eGFR <60 among the study population was 1.6% (95% CI 1.4% to 1.9%). Seventeen per cent (95% CI 16% to 17%) of study participants had eGFR ≥60–<90 mL/min per 1.73 m$^2$ and 82% (95% CI 81% to 82%) had eGFR ≥90 mL/min per 1.73 m$^2$. The prevalences of different categories of eGFR differed by formal education, tobacco consumption, alcohol intake and vegetarianism (table 2). Also, we observed marked differences in the prevalence of eGFR <60 depending on the area, being 1.4% (95% CI 1.1% to 1.8%) in urban areas of Northern India, 1.9 (95% CI 1.4 to 2.6) in rural areas of Northern India, 0.43% (95% CI 0.03% to 0.07%) in urban areas of Southern India and 4.8% (95% CI 3.9% to 5.9%) in the rural area of Southern India. The prevalence ratio of eGFR <60 for rural versus urban residence was higher in participants younger than 50 years (prevalence ratio in age group ≤39=5.5, and prevalence ratio in age group 40–49=5.8) than in older participants (figure 2).

#### Risk factors for lower eGFR and eGFR <60

As expected, age was an important risk factor for reduced eGFR: eGFR was 9.30 mL/min per 1.73 m$^2$ (95% CI –9.51 to –9.09, model adjusted for sex) lower for each additional 10 years of age. Additionally, being male, living in a rural setting and consuming alcohol were associated

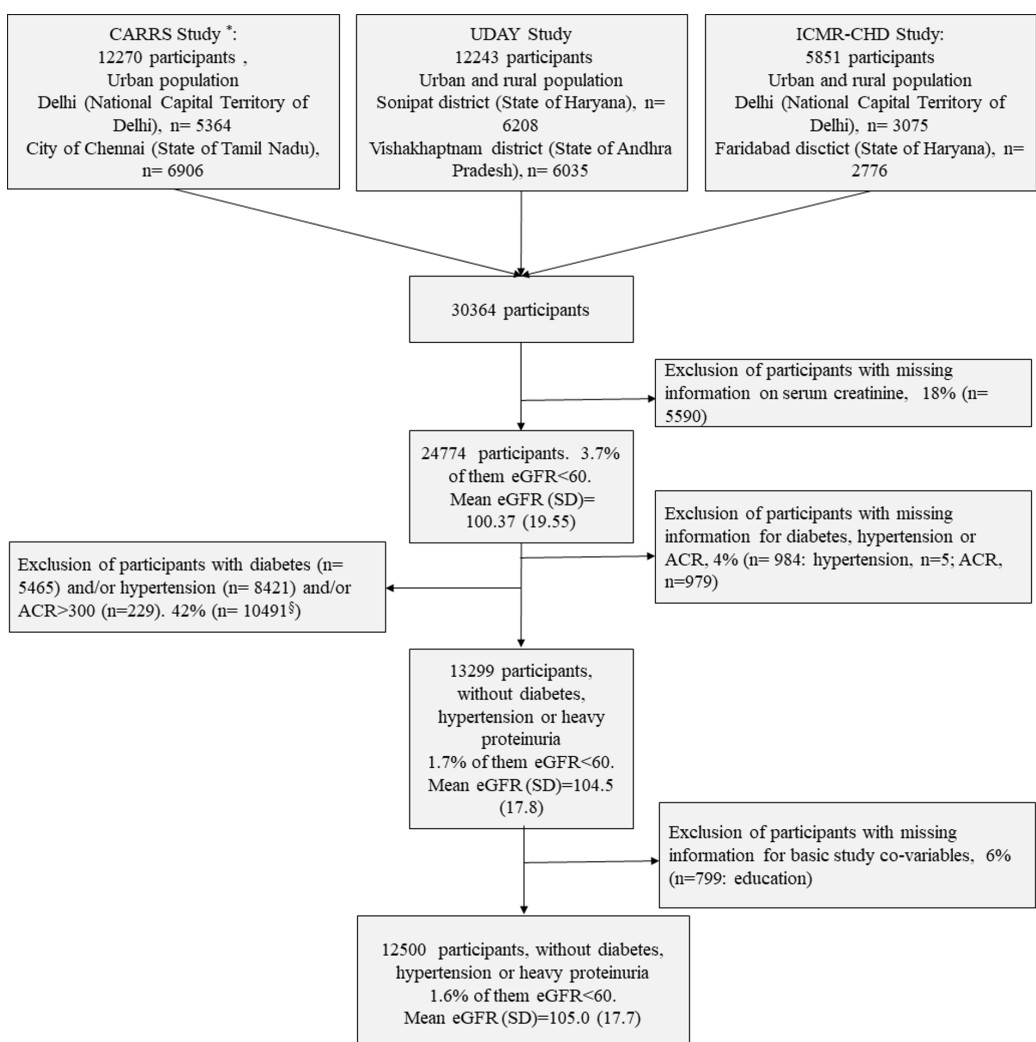

**Figure 1** Study flow chart. ACR, albumin:creatinine ratio; CARRS, Centre for Cardiometabolic Risk Reduction in South Asia; eGFR, estimated glomerular filtration rate; ICMR-CHD, Indian Council of Medical Research Coronary Heart Disease.

with decreased mean eGFR (table 3). Similarly, the odds of eGFR <60 also increased with age (OR per 10 years, adjusted for sex (95% CI)=2.34 (2.12 to 2.59)) and being male, living in a rural setting, living in Southern India and consuming alcohol were also associated with eGFR <60 (table 3). In general, risk factors for decreased mean eGFR and for eGFR <60 were similar for men and women (online supplementary material table S2), but few differences were observed. Regarding mean eGFR, living in Southern India was associated with decreased mean eGFR in men and with increased mean eGFR in women; tobacco consumption was associated with increased mean eGFR in men and with decreased mean eGFR in women; vegetarianism was associated with decreased mean eGFR in women but not in men; and being overweight was associated with decreased mean eGFR but in men but not in women. Regarding risk of eGFR <60, living in Southern India was associated with increased risk of eGFR <60 in men but not in women.

In the multiple regression analyses, decreased mean eGFR remained associated with older age, being male, living in a rural setting and alcohol consumption (table 4). Risk of eGFR <60 remained associated with older age, being male and living in a rural setting, and having no formal education (table 4). We adjusted all the multiple regression models for fat-free mass and vegetarianism to assess the possibility that differences observed between urban and rural participants were due to differences in diet and/or body composition. These adjustments had little effect on the results (table 4).

We observed an interaction between the effects of latitude (North/South) and urban/rural residence in association with reduced eGFR (p value for interaction <0.001). The mean eGFR was lower in rural settings in both Northern and Southern India (controlling for age, sex, education and alcohol intake). However, this decrease was much more marked in Southern India. In Northern India, rural residence, formal education (and

**Table 2** Sociodemographic and anthropometric characteristics of study participants (population without diabetes, hypertension or heavy proteinuria)

| Variable | n (%)* n=12 500 | eGFR mean (SD) | eGFR categories, n (%) [†] | | |
|---|---|---|---|---|---|
| | | | ≥90 | 90–60 | <60 |
| **Sociodemographic** | | | | | |
| Age (years) | | | | | |
| ≤39 | 6121 (49) | 113.8 (14.6) | 5656 (92) | 443 (7) | 22 (0) |
| 40–49 | 3476 (28) | 102.5 (14.2) | 2864 (82) | 572 (16) | 40 (1) |
| 50–59 | 1706 (14) | 93.9 (14.3) | 1163 (68) | 503 (29) | 40 (2) |
| 60–69 | 893 (7) | 85.3 (16.2) | 463 (52) | 368 (41) | 62 (7) |
| ≥70 | 304 (2) | 77.5 (15.1) | 62 (20) | 201 (66) | 41 (13) |
| Sex | | | | | |
| Female | 7066 (57) | 107.9 (17.1) | 6039 (85) | 945 (13) | 82 (1) |
| Male | 5434 (43) | 101.3 (17.9) | 4169 (77) | 1142 (21) | 123 (2) |
| Education (number completed years) | | | | | |
| 0 | 2820 (23) | 100.7 (19.0) | 2165 (77) | 551 (20) | 104 (4) |
| ≤5 | 1709 (14) | 105.9 (17.3) | 1412 (83) | 273 (16) | 24 (1) |
| 6–≤10 | 4817 (39) | 107.2 (16.8) | 4095 (85) | 675 (14) | 47 (1) |
| >10 | 3154 (25) | 105.0 (17.5) | 2536 (80) | 588 (19) | 30 (1) |
| Area [‡] | | | | | |
| Urban | 8494 (68) | 107.8 (16.1) | 7247 (85) | 1171 (14) | 76 (1) |
| Rural | 4006 (32) | 99.0 (18.0) | 2961 (74) | 916 (23) | 129 (3) |
| Latitude [§] | | | | | |
| North | 6263 (50) | 103.0 (17.2) | 4967 (79) | 1197 (19) | 99 (2) |
| South | 6237 (50) | 107.0 (18.1) | 5241 (84) | 890 (14) | 106 (2) |
| **Life-style factors** | | | | | |
| Current tobacco consumption | | | | | |
| No | 9357 (75) | 106.8 (17.3) | 7836 (84) | 1406 (15) | 115 (1) |
| Yes | 3143 (25) | 99.8 (18.1) | 2372 (75) | 681 (22) | 90 (3) |
| Alcohol consumption ever | | | | | |
| No | 10 094 (81) | 105.9 (17.4) | 8362 (83) | 1589 (16) | 143 (1) |
| Yes | 2406 (19) | 101.1 (18.5) | 1846 (77) | 498 (21) | 62 (3) |
| Vegetarian | | | | | |
| No | 7972 (64) | 107.0 (18.0) | 6690 (84) | 1154 (14) | 128 (2) |
| Yes | 4528 (36) | 101.6 (16.6) | 3518 (78) | 933 (21) | 77 (2) |
| **Biological factors** | | | | | |
| Body mass index (kg/m²) | | | | | |
| Underweight (≤18.5) | 5879 (47) | 104.2 (17.9) | 4734 (81) | 1029 (18) | 116 (2) |
| Normal (>18.5–≤25) | 1576 (13) | 104.7 (19.3) | 1283 (81) | 257 (16) | 36 (2) |
| Overweight (>25–≤30) | 3313 (27) | 105.0 (16.9) | 2710 (82) | 568 (17) | 35 (1) |
| Obese (>30) | 1150 (9) | 105.5 (16.4) | 948 (82) | 194 (17) | 8 (1) |
| Missing data | 582 (5) | | 533 (92) | 39 (7) | 10 (2) |
| Fat free mass (kg) | | | | | |
| First tertile (≤37) | 3746 (30) | 106.6 (18.1) | 3146 (84) | 532 (14) | 68 (2) |
| Second tertile (>37 -<45) | 3801 (30) | 105.9 (17.2) | 3145 (83) | 601 (16) | 55 (1) |
| Third tertile (≥45) | 3834 (31) | 102.1 (17.0) | 2981 (78) | 801 (21) | 52 (1) |
| Missing data | 1119 (9) | | 936 (84) | 153 (14) | 30 (3) |

* Percentages in columns.
† Percentages in rows.
‡ Urban areas include Delhi, Chennai and Sonipat district. Rural areas include Sonipat, Vishakhapatnam and Faridabad districts.
§ North areas include Delhi, Sonipat and Faridabad district. South areas include Chennai and Vishakhapatnam districts.
eGFR, estimated glomerular filtration rate.

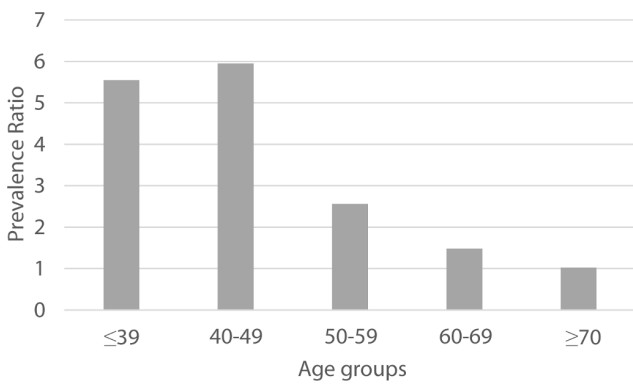

**Figure 2** Prevalence ratio of estimated glomerular filtration rate <60 for rural versus urban residence in different age groups.

duration) and age were the only other risk factor associated with reduced eGFR. In Southern India, being male was also a risk factor for reduced eGFR, whereas formal education was only a risk factor for reduced eGFR among those with more than 10 years of schooling (table 5). We also observed an interaction between the effects of latitude (North/South) and urban/rural residence in association with eGFR <60 (p value likelihood-ratio test for interaction <0.001). In Northern India, eGFR <60 was not associated with urban/rural residence, and older age was the only factor associated with eGFR <60. In Southern India, rural residence was the strongest risk factor for eGFR <60 but older age and lower years of formal education also increased the risk of eGFR <60 (table 5).

### Sensitivity analyses

We performed a sensitivity analysis including those with ACR >300 (but without hypertension or diabetes, n=33) as we were concerned that those with CKDu might develop proteinuria at more advanced CKD stages. However, this did not alter the mean eGFR (mean eGFR among the overall study population=105.0±17.8, mean eGFR in this sensitivity analysis=105.0±17.8), nor the estimated prevalence of eGFR <60 (prevalence among the overall study population=1.6%; prevalence in this sensitivity analysis=1.7%). The findings on risk factors were also similar to the findings from the primary analyses (online supplementary material table S3).

Given concerns about potentially different thresholds to define diabetes and high blood pressure in different ethnic groups,[29 30] we performed a further sensitivity analysis including fasting plasma glucose, HbA1c and systolic blood pressure in the multivariate model (even though there is evidence for both causation and reverse causation between these factors and CKD[31]). Systolic blood pressure and fasting plasma glucose were associated with reduced eGFR in this non diabetic population, but inclusion of these variables did not alter the coefficients for the associations with other risk factors observed in the primary analysis (online supplementary material table S4). HbA1c was associated with eGFR <60 in this non diabetic population but inclusion of this variable did not alter the OR

for other risk factors observed in the primary analysis (online supplementary material table S4). Therefore, although the relationship between subclinical diabetes and impaired kidney function requires further prospective investigation, there is no evidence that the excess risk of low eGFR (ie, lower mean eGFR and higher prevalence of eGFR <60) in rural Southern India is associated with either impaired fasting glucose or higher blood pressure.

### DISCUSSION

We report the distribution of eGFR in people without diabetes, hypertension or heavy proteinuria and estimate the prevalence of CKDu in our study population, including participants from urban and rural settings. This is the first population-based evidence, using standardised methods, which indicates that CKDu is present in India and is not confined to Central America and Sri Lanka. We found that the rural population from Southern India (Vishakhapatnam district) had the highest risk of decreased eGFR (lower mean eGFR and higher prevalence of eGFR <60). Risk factors of decreased eGFR were different between Southern and Northern India. In Southern India, rural residence, older age and being male were risk factors for both lower mean eGFR and eGFR <60; education was associated with decreased risk for eGFR <60 but not with lower mean eGFR. In Northern India, older age was the only risk factor for both lower mean eGFR and eGFR <60; rural residence and years of formal education were associated with lower mean eGFR but not with eGFR <60. In summary, in Southern India, older age, being male and rural residence were the main risk factors for decreased eGFR, whereas in Norther India older age was the main risk factors for decreased eGFR.

As in Central America, the risk of low eGFR was higher in rural settings than in urban settings. This is in concordance with a previous study from Hyderabad (India), that has provided evidence of a higher risk of low eGFR in a rural population compared with urban-migrant and to urban population,[32] and with various studies from other LMICs that have provided evidence of clusters of CKDu among the rural population.[2 3] Exposure to some of the suggested potential risk factors for CKDu such as agricultural work and agrochemical exposure, among others,[33] may be greater in rural settings. Such exposures may also differ between Southern and Northern India, and potentially explain the differences observed between these areas. The associations between urban/rural residence and lower mean eGFR was much more marked in Southern India than in Northern India, and the associations between urban/rural residence and eGFR <60 was only observed in Southern India. The higher prevalence ratio (for eGFR <60) in the working age population compared with older age groups is consistent with the hypothesis that deceased in eGFR could be potentially explained by occupational exposures. The suggestive sex differences may also support this hypothesis. However, we

**Table 3** Associations between sociodemographic and anthropometric characteristics and eGFR and eGFR <60

| Variable | eGFR Coefficient (95% CI) * | eGFR<60 OR (95% CI) * |
|---|---|---|
| Age (years) † | | |
| ≤39 | 0.00 (ref) | 1.00 (ref) |
| 40–49 | −11.08 (−11.68 to −10.47) | 3.15 (1.87 to 5.32) |
| 50–59 | −19.43 (−20.20 to −18.65) | 6.41 (3.80 to 10.83) |
| 60–69 | −27.84 (-28.86 to −26.82) | 19.68 (12.01 to 32.26) |
| ≥70 | −35.04 (−36.71 to −33.37) | 39.23 (22.87 to 67.23) |
| Sex ‡ | | |
| Female | 0.00 (ref) | 1.00 (ref) |
| Male | −3.55 (−4.05 to −3.06) | 1.33 (0.99 to 1.78) |
| Education (number of completed years) | | |
| 0 | 0.00 (ref) | 1.00 (ref) |
| ≤5 | 1.92 (1.09 to 2.76) | 0.41 (0.26 to 0.65) |
| 6–≤10 | 1.27 (0.61 to 1.93) | 0.36 (0.25 to 0.53) |
| >10 | −1.86 (−2.59 to −1.14) | 0.40 (0.26 to 0.62) |
| Area § | | |
| Urban | 0.00 (ref) | 1.00 (ref) |
| Rural | −3.84 (−4.37 to −3.32) | 2.39 (1.78 to 3.22) |
| Latitude ¶ | | |
| North | 0.00 (ref) | 1.00 (ref) |
| South | 0.86 (0.37 to 1.35) | 1.55 (1.16 to 2.07) |
| Current tobacco consumption | | |
| No | 0.00 (ref) | 1.00 (ref) |
| Yes | 0.38 (-−0.26 to 1.02) | 1.39 (1.01 to 1.91) |
| Alcohol consumption ever | | |
| No | 0.00 (ref) | 1.00 (ref) |
| Yes | −0.81 (−1.55 to −0.08) | 1.57 (1.09 to 2.27) |
| Vegetarian | | |
| No | 0.00 (ref) | 1.00 (ref) |
| Yes | −0.99 (−1.50 to −0.47) | 0.65 (0.48 to 0.88) |
| Body mass index (kg/m$^2$) | | |
| Underweight (≤18.5) | 2.96 (2.20 to 3.73) | 0.81 (0.55 to 1.20) |
| Normal (>18.5–≤25) | 0.00 (ref) | 1.00 (ref) |
| Overweight (>25–≤30) | −0.75 (−1.34 to −0.16) | 0.68 (0.46 to 1.01) |
| Obese (>30) | −0.71 (−1.59 to 0.17) | 0.47 (0.23 to 0.98) |
| Fat-free mass (kg) | | |
| 1st tertile (≤37) | 0.00 (ref) | 1.00 (ref) |
| 2nd tertile (>37–<45) | −0.91 (−1.54 to −0.28) | 0.69 (0.47 to 1.03) |
| 3rd tertile (≥45) | −3.90 (−4.77 to −3.04) | 0.49 (0.31 to 0.80) |

*Adjusted for age and sex.
†Adjusted just for sex.
‡Adjusted just for age.
§Urban areas include Delhi, Chennai and Sonipat district. Rural areas include Sonipat, Vishakhapatnam and Faridabad districts.
¶ North areas include Delhi, Sonipat and Faridabad district. South areas include Chennai and Vishakhapatnam districts.
eGFR, estimated glomerular filtration rate.

**Table 4** Multiple regression analyses of sociodemographic characteristics associated with with eGFR and eGFR <6)

| Variable | eGFR coefficient (95% CI) | | | eGFR <60 OR (95% CI) | | |
|---|---|---|---|---|---|---|
| | Model 1 [*] | Model 2 [†] | Model 3 [‡] | Model 1 [*] | Model 2 [†] | Model 3 [‡] |
| Area [§] | | | | | | |
| Urban | 0.00 (ref) | 0.00 (ref) | 0.00 (ref) | 1.00 (ref) | 1.00 (ref) | 1.00 (ref) |
| Rural | −4.57 (−5.13 to 4.02) | −3.94 (−4.53 to 3.36) | −4.10 (−4.70 to −3.51) | 1.99 (1.43 to 2.76) | 1.61 (1.12 to 2.30) | 1.65 (1.14 to 2.37) |
| Latitude [¶] | | | | | | |
| North | 0.00 (ref) | 0.00 (ref) | 0.00 (ref) | 1.00 (ref) | 1.00 (ref) | 1.00 (ref) |
| South | 0.31 (−0.18 to 0.80) | −0.10 (−0.61 to 0.41) | 0.26 (−0.37 to 0.89) | 1.33 (0.98 to 1.81) | 1.60 (1.14 to 2.32) | 1.33 (0.86 to 2.04) |
| Education (number of completed years) | | | | | | |
| 0 | 0.00 (ref) | 0.00 (ref) | 0.00 (ref) | 1.00 (ref) | 1.00 (ref) | 1.00 (ref) |
| ≤5 | 0.94 (0.01 to 1.77) | 1.16 (0.30 to 2.02) | 1.18 (0.32 to 2.04) | 0.50 (0.31 to 0.80) | 0.44 (0.26 to 0.74) | 0.45 (0.26 to 0.75) |
| 6–≤10 | 0.04 (−0.64 to 0.72) | 0.21 (−0.49 to 0.91) | 0.21 (−0.50 to 0.92) | 0.50 (0.34 to 0.75) | 0.38 (0.24 to 0.60) | 0.39 (0.25 to 0.62) |
| >10 | −3.81 (−4.6 to 3.0) | −3.81 (−4.60 to 3.02) | −3.78 (−4.59 to −2.97) | 0.68 (0.42 to 1.11) | 0.61 (0.36 to 1.03) | 0.65 (0.38 to 1.11) |
| Alcohol consumption ever | | | | | | |
| No | 0.00 (ref) | 0.00 (ref) | 0.00 (ref) | 1.00 (ref) | 1.00 (ref) | 1.00 (ref) |
| Yes | −0.85 (−1.58 to 0.12) | −0.69 (−1.47 to 0.08) | −0.63 (−1.41 to 0.15) | 1.28 (0.88 to 1.87) | 1.18 (0.78 to 1.79) | 1.15 (0.76 to 1.74) |
| Sex | | | | | | |
| Female | 0.00 (ref) | 0.00 (ref) | 0.00 (ref) | 1.00 (ref) | 1.00 (ref) | 1.00 (ref) |
| Male | −2.85 (−3.44 to 2.25) | −3.00 (−3.62 to 2.38) | −2.52 (−3.18 to −1.86) | 1.39 (0.96 to 2.01) | 1.49 (1.00 to 2.21) | 1.50 (0.97 to 2.31) |
| Age (per 10 years) | −9.10 (−9.32 to 8.88) | −9.09 (−9.32 to 8.86) | −9.15 (−9.38 to −8.91) | 2.21 (1.98 to 2.47) | 2.25 (2.00 to 2.55) | 2.27 (2.00 to 2.57) |
| Fat-free mass (kg) | | | −0.04 (−0.06 to −0.02) | | | 1.0 (0.98 to 1.02) |
| Vegetarian | | | | | | |
| No | | | 0.00 (ref) | | | 1.00 (ref) |
| Yes | | | 0.66 (−0.03 to 1.35) | | | 0.74 (0.47 to 1.18) |

*Model 1 included the following variables: area, latitude, education, alcohol consumption, sex and age; n=12 500.
†Model 2 included the same variables than model 1. Participants with missing information on fat-free mass were excluded from the analysis, n=11 381.
‡Model 3 included the same variables than model 1 plus fat-free mass and vegetarianism, n=11 381.
§Urban areas include Delhi, Chennai and Sonipat district. Rural areas include Sonipat, Vishakhapatnam and Faridabad districts.
¶North areas include Delhi, Sonipat and Faridabad district. South areas include Chennai and Vishakhapatnam districts.
eGFR, estimated glomerular filtration rate.

**Table 5** Multivariate analysis of sociodemographic characteristics associated with eGFR and with eGFR <60 according to latitude*

| Variables | eGFR (n=12500) | | eGFR<60(n=12500) | |
| | North (n=6263) [†] | South (n=6237) [‡] | North (n=6263) [†] | South (n=6237) [‡] |
| | Coefficient (95% CI) | Coefficient (95% CI) | OR (95% CI) | OR (95% CI) |
|---|---|---|---|---|
| Area [§] | | | | |
| Urban | 0.00 (ref) | 0.00 (ref) | 1.00 (ref) | 1.00 (ref) |
| Rural | −1.42 (−2.15 to 0.70) | −7.90 (−8.81 to 7.00) | 0.88 (0.57 to 1.37) | 4.68 (2.50 to 8.77) |
| Education (number of completed years) | | | | |
| 0 | 0.00 (ref) | 0.00 (ref) | 1.00 (ref) | 1.00 (ref)* |
| ≤5 | −1.32 (−2.58 to 0.05) | 1.05 (−0.06 to 2.16) | 1.16 (0.57 to 2.35) | 0.40 (0.20 to 0.80) |
| 6–≤10 | −3.50 (−4.48 to 2.52) | 0.28 (−0.74 to 1.30) | 1.34 (0.74 to 2.41) | 0.35 (0.16 to 0.74) |
| >10 | −6.93 (−7.97 to 5.89) | −2.85 (−4.03 to 1.67) | 1.34 (0.69 to 2.58) | 0.61 (0.24 to 1.57) |
| Alcohol consumption ever | | | | |
| No | 0.00 (ref) | 0.00 (ref) | 1.00 (ref) | 1.00 (ref) |
| Yes | −0.54 (−1.55 to 0.47) | −0.06 (−1.11 to 0.99) | 1.09 (0.62 to 1.92) | 1.36 (0.74 to 2.17) |
| Sex | | | | |
| Female | 0.00 (ref) | 0.00 (ref) | 1.00 (ref) | 1.00 (ref) |
| Male | −0.17 (−0.96 to 0.63) | −5.40 (−6.29 to 4.51) | 0.97 (0.59 to 1.59) | 1.58 (0.91 to 2.75) |
| Age (per 10 years) | −9.26 (−9.55 to 8.97) | −8.96 (−9.28 to 8.64) | 2.51 (2.15 to 2.93) | 2.10 (1.77 to 2.50) |

*Likelihood ratio test for linear trend <0.05, OR (95% CI)=0.68 (0.51 to 0.91).
†North areas include Delhi, Sonipat and Faridabad district.
‡South areas include Chennai and Vishakhapatnam districts.
§Urban areas include Delhi, Chennai and Sonipat district. Rural areas include Sonipat, Vishakhapatnam and Faridabad districts.
eGFR, estimated glomerular filtration rate.

did not have detailed data on occupation that allowed us to explore these associations in greater detail.

The higher risk of low eGFR in Southern India (Chennai and Vishakhapatnam districts) observed in our study is consistent with the clusters of CKDu cases previously reported in the Southern Indian states of Andhra Pradesh and Odisha.[11 18 19] Vishakhapatnam district (state of Andhra Pradesh) and Chennai district (state of Tamil Nadu) have a similar climate than these areas where CKDu clusters have previously reported.[34] In these districts, mean temperatures range from 18°C to 37°C and rainfall occurs mainly between June and December.[35] On the other hand, sites from Norther India included in the study (Delhi (National Capital Territory of Delhi)), Sonipat and Faridabad (Haryana state)), have a different climate. In these districts mean temperature ranges from 8°C to 39°C and precipitation occurs mainly between July and August.[35] A previous study conducted in Costa Rica found a spatial correlation between rates of CKD mortality and temperature and rainfall.[13]

About 5% of the rural population of Vishakhapatnam (Andra Pradesh, Southern India) without diabetes, hypertension or proteinuria had eGFR <60. This figure is almost as high as the prevalence observed in the USA (ie, 6.7%) including people with diabetes, hypertension or proteinuria.[36] Moreover, the estimates of GFR in our study are likely to be underestimated. The CKD-EPI equation has been standardised for the white and Afro-American population,[23] but its validity for other ethnic groups has been questioned.[37 38] Previous studies using CKD-EPI equation to estimate GFR in Indian populations reported mean eGFR values similar to the mean eGFR reported in our study (ie, $104.9\pm25.52\,\text{mL/min}/1.73\,\text{m}^2$).[39] However, two studies conducted among healthy kidney donors in India (population similar to those included in this analysis) have reported mean (measured) GFR between 81.4 and 95.5 mL/min per $1.73\,\text{m}^2$,[40 41] suggesting that the CKD-EPI equation substantially overestimates eGFR in the Indian population. Therefore, the prevalence of eGFR <60 observed in this study is likely to be substantially underestimated (although this is unlikely to have biased the internal comparisons, eg, between urban and rural settings). The use of a conservative definition of the population susceptible to CKDu, may have also underestimated the prevalence of eGFR <60 in our study, as the population with diabetes, hypertension or glomerular disease may also have reduced eGFR due to other ('unknown') causes. To estimate the actual prevalence of reduced eGFR, future studies should include validated methods to estimate GFR in the Indian population. We were concerned that the validity of CKD-EPI among the Indian population may be also compromised by differences in muscular mass and meat consumption between

population groups within India. We adjusted the analyses for fat free mass and vegetarianism, but this did not alter the results, suggesting no confounding effect by these variables.

Our study has at least three potential limitations. First, we only had one measure of eGFR, and therefore we could not differentiate acute kidney injury (AKI) from CKD. This is a common limitation in epidemiological studies, as it is challenging to obtain more than one measure of eGFR at least 3 months apart in large population-based investigations. Therefore, we may have misclassified some cases of AKI as reduced eGFR, and therefore overestimate the prevalence of this condition. Nevertheless, there is no a priori reason to think that potential misclassification was different according to the evaluated risks factors. Second, the three population-based studies included in this analysis used different sampling strategies. CARRS and UDAY studies included only one man and one woman from all the eligible participants of selected households, whereas ICMR-CHD included all eligible adults from each selected household. This could have slightly biased our results (including our prevalence estimates) if risk factors potentially associated with CKDu were different between households inhabited only by a man and a women or by extended families. Third, information on other potential risk factors for CKDu, such as infections by *Leptospora* or hantavirus infection, or use of non-steroidal anti-inflammatory drugs was not available.

The main strengths of the study are the use of a random selection of population-based participants and a large sample size including participants from different areas of India (urban and rural, and Northern and Southern India). Moreover, we used the definitions proposed in DRGREE study,[42] that aims to allow international comparisons of CKDu prevalence and help in the description of risk factors and in identifying the causes and mechanisms leading to CKDu.

In conclusion, our findings indicate that reduced eGFR, consistent with the definition of CKDu, is common in rural settings of Southern India (Vishakhapatnam district). This results support the hypothesis that the epidemic of CKDu, initially described in agricultural communities of Central America and Sri Lanka, may be common in other rural communities of tropical/subtropical countries. This has important implications for global health, since it indicates that CKDu may have a substantial public health burden globally that has been previously unrecognised. Population-based studies in other tropical/subtropical countries are required to assess the global patterns of burden of disease from CKDu.[42]

**Author affiliations**
[1]ISGlobal, Barcelona, Spain
[2]Universitat Pompeu Fabra (UPF), Barcelona, Spain
[3]CIBER Epidemiología y Salud Pública (CIBERESP), Madrid, Spain
[4]Department of Medical Statistics, London School of Hygiene and Tropical Medicine, London, UK
[5]Public Health Foundation of India, Gurgaon, Haryana, India
[6]Centre for Control of Chronic Conditions (4Cs), New Delhi, Haryana, India
[7]Stanford University School of Medicine, Stanford, CA, USA
[8]LalsIa Network, Ada, Michigan, USA
[9]Occupational and Environmental Medicine, Sahlgrenska Academy, Gothenburg University, Gothenburg, Sweden
[10]Occupational and Environmental Medicine, Lund University, Lund, Sweden
[11]Centre for Community Medicine, All India Institute of Medical Sciences, New Delhi, Haryana, India
[12]Diabetes Research, Madras Diabetes Research Foundation, Chennai, India
[13]Dr.Mohan's Diabetes Specialities Centre, Chennai, India
[14]Department of Epidemiology and Population Health, London School of Hygiene and Tropical Medicine, London, UK
[15]Department of Endocrinology and Metabolism, All India Institute of Medical Sciences, New Delhi, India
[16]Emory Global Diabetes Research Center, Rollins School of Public Health, Emory University, Atlanta, GA, USA
[17]Centre for Global NCDs, London School of Hygiene and Tropical Medicine, London, UK
[18]Centre for Nephrology, University College London Medical School, London, UK

**Correction notice** This article has been corrected since it first published. After the publication of this article, the authors noticed that the map of India showing the study sites needed correction. The borders of India shown in the map were inaccurate and the authors have therefore decided to withdraw this map from the publication. The article has therefore been republished without the original figure 1. Any readers with an academic interest in the information originally displayed in the figure may contact the authors directly in relation to this information.

**Acknowledgements** We thank Manolis Kogevinas for his comments on the advanced version of the manuscript.

**Contributors** CO-G, BC, NP and DP designed the work; RS, SA, SG, RG, AK, SM, VM, PPA, NT and KMN collected the data; CO-G and DK conducted the analysis of the data; CO-G, RS, SA, JG, KJ, DN, SM, KMN, NP, BC and DP interpreted the data of the work. CO-G, RS, BC, and NP drafted the manuscript; RS, SA, SG, JG, RG, KJ, DK, AK, SM, VM, DN, PPA, NT, KMN and DP revised the manuscript for important intellectual content, provided comments and suggested revisions. All authors approved the final version for publication.

**Funding** This work was supported in part by grant MR/P02386X/1 from the United Kingdom Medical Research Council under the Global Challenges Research Fund. It was also supported by grants from the Colt Foundation and the La Isla Foundation. The CARRS study was funded with federal funds from the National Heart, Lung, and Blood Institute, National Institutes of Health, under Contract No. HHSN2682009900026C. UDAY study was funded by Eli Lilly Foundation. ICMR-CHD study was funded by the Indian Council Medical Research (ICMR). The Centre for Global NCDs is supported by the Wellcome Trust Institutional Strategic Support Fund (097834/Z/11/B). CO-G was supported by a Sara Borrell postdoctoral fellowship awarded from the Carlos III National Institute of Health, Spain (CD13/00072).

**Competing interests** None declared.

**Patient consent for publication** Not required.

**Ethics approval** Participants from CARRS, UDAY and ICMR-CHD studies provided informed consent prior to participation. The three studies obtained ethical clearance from the corresponding institutions.

**Provenance and peer review** Not commissioned; externally peer reviewed.

**Data sharing statement** The datasets used and/or analysed during the current study are available from Public Health Foundation of India (PHFI) on reasonable request. Interested investigators should contact PHFI. Computing code can be obtained from the corresponding author.

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
