## [Reviewer comments · BMJ Open]

This paper was submitted to a another journal from BMJ but declined for publication following peer review. The authors addressed the reviewers' comments and submitted the revised paper to BMJ Open. The paper was subsequently accepted for publication at BMJ Open.

(This paper received three reviews from its previous journal but only two reviewers agreed to published their review.)

ARTICLE DETAILS

TITLE (PROVISIONAL)	Prevalence of and risk factors for chronic kidney disease of unknown aetiology in India: secondary data analysis of three population-based cross-sectional studies
AUTHORS	O'Callaghan Gordo, Cristina; Shivashankar, Roopa; Anand, Shuchi; Ghosh, Shreeparna; Glaser, Jason; Gupta, Ruby; Jakobsson, Kristina; Kondal, Dimple; Krishnan, Anand; Mohan, Sailesh; Mohan, V; Nitsch, Dorothea; PA, Praveen; Tandon, Nikhil; Narayan, K; Pearce, Neil; Caplin, Ben; Prabhakaran, Dorairaj

VERSION 1 – REVIEW

REVIEWER	Hooi Lai Seong Sultanah Aminah Hospital Johor Bahru Malaysia
REVIEW RETURNED	14-May-2018

GENERAL COMMENTS	The UDAY study cannot be found and so its methodology cannot be evaluated. Had downloaded the ICMR study as an abstract but have gotten both the Anand and Nair papers (for CARRS). In figure 1 the CARRS study numbers differ from the Anand paper by 1 (12,270 vs 12,271). The numbers for the UDAY study cannot be verified as there is no paper published. For the ICMR study the Prabhakaran abstract says for survey 2 (2010 – 2012) the number for Delhi was 2,052 and for Haryana it was 1,917 – these 2 numbers do not tally with those on figure 1 (3,075 and 2,776 respectively). Did the authors have patient level data for the final 12,500 participants or were they supplied with tables separately by each of the authors of the 3 studies. From the Anand paper (CARRS) there was no central laboratory for estimating serum creatinine although each local lab subscribed to a quality assurance program. In table 1 it is stated the laboratory was different between Delhi and Chennai (PHFI and MDRF). For the other 2 studies it was not possible to ascertain whether it was a central lab or not and whether each used a standard method traceable to IDMS for serum creatinine. For ACR (albumin creatinine ratio) the method was unknown except in the CARRS study. As these 2 parameters form the basis for eGFR and
---

	proteinuria definition in this paper this should be clarified further. Were the definitions of diabetes and hypertension uniform throughout all 3 studies using the same instruments e.g. sphygmomanometers; was height and weight measured with the same scales? From the source papers could not find the methodology for fat free mass and what meter was used for 3 studies. The results show there were more women than men 57: 43 (table 2) in the cohort. In table S1 there is an error n = 24,774 and not 12,500. In table 3 living in South India was not associated with decreased eGFR (cf line 181 of manuscript). From table S3 living in the South was not a factor for decreased eGFR (cf line 184 and 185 of the manuscript). In table 4 being male was not a negative factor for mean eGFR in Model 2 and Model 3 (cf line 186 of the manuscript). For eGFR < 60 in table 4, being male does not seem to be a risk factor, and the 3 models with age on it seem too uniform with the same result and the same CI i.e. 1.1 (1.1-1.1) (cf line 187 and 188 of the manuscript). It is possible the "male" calculations are wrong? There is something wrong with table 5 on the right-hand side about eGFR < 60 (last 2 columns). I think the rows were inadvertently shifted down 1 row each such that there is no data for "age" right at the bottom for both North and South India – this was from eyeballing it for quite a long time. For table S3 being male seem to be a risk factor for low eGFR unlike table 4. The result had changed quite a bit although only 33 participants had been added.
--	---

REVIEWER	Maurizio Gallieni University of Milano, Italy
REVIEW RETURNED	19-May-2018

GENERAL COMMENTS	This article extends the knowledge of a still undefined clinical entity, i.e. the occurrence of chronic kidney disease in tropical and subtropical areas, whose cause is still unknown. Chronic underhydration might be an intuitive cause (cyclical dehydration-induced renal injury), but other causes are possible, including toxic nephropathies. The three studied populations are large and well defined. Although the studies is a secondary analysis of previous studies designed with a different aim, the results appear to be robust and support the conclusions. I applaud the efforts of deepening our knowledge of an "orphan" disease, which affects many people and families in the affected areas. I suggest acceptance for publication.
--

REVIEWER	Toshiki Moriyama Toshiki Moriyama
REVIEW RETURNED	03-Jul-2018

GENERAL COMMENTS	The present manuscript entitled "Prevalence of and risk factors for chronic kidney disease of unknown aetiology in India: secondary data analysis of three population-based cross-sectional studies" by O'Callaghan-Gordo C. et al. reported that rural area, lower education level, higher age, higher HbA1c was associated with prevalence of CKD defined as eGFR <60 mL/min/1.73 m² in the subjects without diabetes, hypertension, and overt proteinuria, using 3 large population-based studies. I think that the findings of large regional differences in prevalence of CKD in India probably provides important information to make a strategy for prevention of CKD. However, this manuscript was so descriptive that it was hard to understand what was the key finding of the present study. I would like to advise the authors to reorganize this manuscript to stress what was the main novel findings. One of the candidates might be effect modification between area and education because it was very difficult to assess a clinical impact of education on the prevalence of CKD in western countries and its impact was dependent on areas (Table 5). The authors should reorganize their manuscript to make their research question clear. 1) Because the majority of the subject was within the normal range of eGFR, I do not think the multivariable-adjusted linear regression models with eGFR as a dependent variable had no clinical value. For example, in Table 2, we should pay our attention not to mean eGFR of education (0, ≤5, 6-10, and >10: 100.7, 105, 9, 107, 2, and 105.0, respectively), but to the prevalence of eGFR <60 (5%, 1%, 1%, and 1%, respectively). I think that eGFR was not appropriate as an dependent variable in this manuscript, which should be supplementary if the authors should to show. 2) To provide the clinical characteristics of patients included in the present study in more details, the main purposes of CARRS study, UDAY study, and ICMR-CHD study. 3) Clinical characteristics should be shown after stratified by the key exposure variable, instead of Table 2. How did the authors define rural and urban area and north and south latitude? A map of study area help the readers know the studies places. 4) What was mutual adjustment in Table 4? All covariates should be described in the footnotes of the tables. 5) The authors should pay the unit of each covariates included in the multivariable-adjusted model. Coefficients or odds ratio of age (per 1 year) was ridiculous. A risk elevation of a 1-year difference has no clinical value. Unit of age should be "per 10 year" to estimate the clinically relevant coefficients and odds ratios.
---

VERSION 1 – AUTHOR RESPONSE

Reviewer: 1

Reviewer Name: Hooi Lai Seong

Institution and Country: Sultanah Aminah Hospital, Johor Bahru, Malaysia

Please state any competing interests or state 'None declared': None declared

Please leave your comments for the authors below

The UDAY study cannot be found and so its methodology cannot be evaluated. Had downloaded the ICMR study as an abstract but have gotten both the Anand and Nair papers (for CARRS).

In figure 1 the CARRS study numbers differ from the Anand paper by 1 (12,270 vs 12,271).

Response: One of the participants from the CARRS study was a transgender person. In the current analysis, both biological sex and gender were relevant for the association under study, and for this reason we excluded this person. In the revised version of the manuscript we have added this information as a footnote in the figure (now figure 2): “The original sample size in the CARRS study is 12271, one transgender person was excluded for the current analysis”

The numbers for the UDAY study cannot be verified as there is no paper published.

Response: The paper is now published: <https://bmjopen.bmj.com/content/8/6/e015919>

For the ICMR study the Prabhakaran abstract says for survey 2 (2010 – 2012) the number for Delhi was 2,052 and for Haryana it was 1,917 – these 2 numbers do not tally with those on figure 1 (3,075 and 2,776 respectively).

Response: Prabhakaran et al (<https://www.ncbi.nlm.nih.gov/pubmed/28411147>) compared the data from the two ICMR surveys: survey 1 was conducted between 1991-1994 and recruited individuals aged 35-64 years. Survey 2 was conducted between 2010-2012 and recruited individuals older than 30 years. For the comparison of the two surveys, Prabhakaran et al, restricted the analysis to the 35-64 years age group, whereas for the current we included all individual >30 years as we only used data from survey 2 (2010-2012).

Did the authors have patient level data for the final 12,500 participants or were they supplied with tables separately by each of the authors of the 3 studies.

Response: Yes, we did have access to patient level data from the three studies. We homogenized the information from the three studies and merged the data sets from the three studies. We used this homogenized data set to conduct all the analysis presented in the current manuscript. We have included this information in the revised version of the manuscript: “Data from the three studies were homogenized and merged in a single data set.”

From the Anand paper (CARRS) there was no central laboratory for estimating serum creatinine although each local lab subscribed to a quality assurance program. In table 1 it is stated the laboratory was different between Delhi and Chennai (PHFI and MDRF). For the other 2 studies it was not possible to ascertain whether it was a central lab or not and whether each used a standard method traceable to IDMS for serum creatinine. For ACR (albumin creatinine ratio) the method was unknown except in the CARRS study. As these 2 parameters form the basis for eGFR and proteinuria definition in this paper this should be clarified further.

Response: All samples from UDAY, ICMR-CHD, and samples from CARRS from Delhi were analysed at PHFI laboratory. Samples from CARRS from Chennai were analysed at MDRF laboratory following the same methodology and protocols. Both PHFI and MDRF laboratories used the same methodology and protocols to estimate serum creatinine and participated in Randox International Quality Assurance Scheme (RIQAS) for clinical chemistry and HbA1c during the entire study periods.

So, analyses serum creatinine for UDAY, ICMR-CHD, and samples from CARRS from Delhi were done at PHFI laboratory, and for CARRS from Chennai were done at MDRF laboratory.

Microalbuminuria was measured using immune turbidimetric method in the three studies.

As for creatinine, samples from UDAY, ICMR-CHD, and samples from CARRS from Delhi were analysed at PHFI laboratory. Samples from CARRS from Chennai were analysed at MDRF laboratory. ACR was measured by dividing levels of albuminuria (mg/dl) by levels of creatinine (g/dl) for all participants of the three studies.

We have included this information in the new version of the manuscript: “Glucose levels were measured using hexokinase/kinetic methods, HbA1c using high-performance liquid chromatography, serum creatinine using the rate-blanked and compensated kinetic Jaffe method, traceable to isotope dilution mass spectrometry, and albuminuria using immune turbidimetric method (Nair et al. 2012). Samples from UDAY, ICMR-CHD, and samples from CARRS from Delhi were analysed at Public Health Foundation of India (PHFI) laboratory and samples from CARRS from Chennai were analysed

at Madras Diabetes Research Foundation (MDRF) laboratory. Both PHFI and MDRF laboratories used the same methodologies and protocols to analyse the samples and participated in Randox International Quality Assurance Scheme (RIQAS) for clinical chemistry and HbA1c during the entire study periods.”

Were the definitions of diabetes and hypertension uniform throughout all 3 studies using the same instruments e.g. sphygmomanometers; was height and weight measured with the same scales? From the source papers could not find the methodology for fat free mass and what meter was used for 3 studies.

Response: For participants of the three studies we had information of plasma fasting glucose, glycated haemoglobin A1c and participant self-reported diabetes. We used this information to generate a common definition of diabetes for the three studies: “We classified participants as having: diabetes, if plasma fasting glucose was ≥ 126 mg/dl, or glycated haemoglobin A1c (HbA1c) was $\geq 6.5\%$, or the participant self-reported diabetes”. Similarly, we had information of systolic and diastolic blood pressure and participant self-reported hypertension. Based on this information we generated a common definition of hypertension for the three studies: “We classified participants as having (...) hypertension, if systolic blood pressure was ≥ 140 mm Hg, or diastolic blood pressure was ≥ 90 mm Hg, or the participant self-reported hypertension”.

In CARRS and ICMR-CHD studies the same models of stadiometers, weighing scales and sphygmomanometers were used. In UDAY study also the same stadiometers were used, whereas the weighing scales and sphygmomanometers used were slightly different (same brand but newer model). All instruments were calibrated before each study, and no re-calibration was needed during the duration of the different studies. We have included this information in the new version of the manuscript as well as details on how fat free mass was estimated:

“Weight, height and body composition were measured using stadiometers (SECA 214 in the three studies) and electronic bioimpedance measuring instruments (Tanita BC 418 in CARRS and ICMR-CHD studies, and Tanita BC 601 in UDAY study). Body mass index (BMI, kg/m²) was calculated and categorized (≤ 18.5 : underweight; >18.5 - ≤ 25 : normal weight; >25 - ≤ 30 : overweight; >30 : obese) and fat free mass was derived from bioelectric impedance analysis (BIA). In CARRS and ICMR-CHD studies, fat free mass (Kg) was directly measured as previously described (Patel 2017), whereas in UDAY study, fat free mass was estimated from the percentage of total body fat. To estimate total fat free mass from the percentage of body fat, we calculated the amount of total body fat by multiplying the percentage of body fat by the weight of the participant, and from that value we estimated the amount of fat free mass by subtracting the weight of total body fat from the total weight of the participant. Blood pressure was measured using electronic sphygmomanometers (OMRON (HEM-7080) in CARRS and ICMR-CHD studies, and OMRON (HEM 7200) in UDAY study), as previously reported (Nair et al. 2012; Anand et al. 2015). Stadiometers, electronic weighing scales, and electronic sphygmomanometers were calibrated before each study, and no re-calibration was needed during the duration of different studies.”

The results show there were more women than men 57: 43 (table 2) in the cohort.

Response: Yes, there were more women than men in our study sample. Having differences in the proportion of women and men in our study sample is not surprising, as the sampling strategy did not aim at having the same percentage of man and women, but to be representative of the population under study (population without diabetes, hypertension or heavy proteinuria).

Briefly, the urban ICMR-CHD sample was selected using a multi-stage cluster random sampling technique, being wards (larger municipal divisions) the primary sampling units and census enumeration blocks the secondary sampling units. The rural sample was selected using population level sampling frame and a simple random sampling was used to select households. In ICMR-CHD study all adults from selected households were recruited into the study. Selection of the study participants in CARRS and UDAY was done following different strategies. In CARRS, households were selected using the same multi-stage cluster random sampling approach than urban ICMR-CHD sample (wards - census enumeration blocks - households). In the UDAY study, households were also

selected using a multi-stage cluster random sampling technique. The primary sampling units were census enumeration blocks in urban areas and villages in rural areas. Both, in CARRS and UDAY studies, from each household 1 male and 1 female were selected using the WHO Kish method. In households where only eligible men or only eligible women were present, we selected just one adult. This information was not included in the previous version of the manuscript and these could have created confusion. We have included it as a footnote of table 1 "In households where only eligible men or only eligible women were present, we selected just one adult".

With these sampling strategies, the proportion of men and women in our study sample might reflect the proportion if men and women in the population under study (population without diabetes, hypertension or heavy proteinuria). From our results seem that there were slightly more women than men fulfilling the study criteria.

In table S1 there is an error $n = 24,774$ and not 12,500.

Response: Thank you for noticing this mistake. We have corrected it in the reviewed version of the manuscript.

In table 3 living in South India was not associated with decreased eGFR (cf line 181 of manuscript).

Response: We have corrected this in the reviewed version of the manuscript: "Additionally, being male, living in a rural setting, and consuming alcohol were associated with decreased mean eGFR (Table 3)."

From table S3 living in the South was not a factor for decreased eGFR (cf line 184 and 185 of the manuscript).

Response: Lines 184 and 185 refer to table S2 ("Associations between sociodemographic and anthropometric characteristics and estimated glomerular filtration rate (eGFR) and eGFR<60 by sex"). We have included more information about table S2 in the text and commented that living in South India was a risk factor for decreased mean eGFR and increase risk for eGFR < 60 in men but not in women: "In general, risk factors for decreased mean eGFR and for eGFR<60 were similar for men and women (supplementary material, Table S2), but few differences were observed. Regarding mean eGFR, living in Southern India was associated with decreased mean eGFR in men and with increased mean eGFR in women; tobacco consumption was associated with increased mean eGFR in men and with decreased mean eGFR in women; vegetarianism was associated with decreased mean eGFR in women but not in men; and being overweight was associated with decreased mean eGFR but in men but not in women. Regarding risk of eGFR<60, living in Southern India was associated with increased risk of eGFR<60 in men but not in women."

Results from table S3 ("Multiple regression analysis of sociodemographic characteristics associated with eGFR and eGFR<60 including study participants with proteinuria (but without diabetes or hypertension), $n=12533$ ") are very similar to results from the main analysis (table 5). In both tables, living in South India was associated with increased risk for eGFR<60, but not with decreased mean eGFR.

In table 4 being male was not a negative factor for mean eGFR in Model 2 and Model 3 (cf line 186 of the manuscript). For eGFR < 60 in table 4, being male does not seem to be a risk factor, and the 3 models with age on it seem too uniform with the same result and the same CI i.e. 1.1 (1.1-1.1) (cf line 187 and 188 of the manuscript). It is possible the "male" calculations are wrong?

Response: There was a mistake in tables 4 and 5 regarding the variable sex. Models presented in these tables, considered "male" as the baseline category, whereas for the other tables of the paper "female" was considered the baseline category. We have corrected this mistake and revised all tables in the new version of the manuscript. According the new version of the table, being male is a risk factor for decreased mean eGFR in all models considered and for increased risk of eGFR<60.

Results of the variable age are indeed very uniform in the three models, but we have included two decimals to show differences between models. We have done this for all tables presented.

There is something wrong with table 5 on the right-hand side about eGFR < 60 (last 2 columns). I think the rows were inadvertently shifted down 1 row each such that there is no data for "age" right at the bottom for both North and South India – this was from eyeballing it for quite a long time.

Response: The reviewer is right. We have corrected table 5 in the new version of the manuscript.

For table S3 being male seem to be a risk factor for low eGFR unlike table 4. The result had changed quite a bit although only 33 participants had been added.

Response: As commented above, there was an error in table 4 regarding the variable sex. In the revised version of the manuscript, in both tables 4 and S3 being male is a risk factor for low eGFR.

Reviewer: 2

Reviewer Name: Maurizio Gallieni

Institution and Country: University of Milano, Italy

Please state any competing interests or state 'None declared': None declared

Please leave your comments for the authors below

This article extends the knowledge of a still undefined clinical entity, i.e. the occurrence of chronic kidney disease in tropical and subtropical areas, whose cause is still unknown. Chronic underhydration might be an intuitive cause (cyclical dehydration-induced renal injury), but other causes are possible, including toxic nephropathies.

The three studied populations are large and well defined. Although the studies is a secondary analysis of previous studies designed with a different aim, the results appear to be robust and support the conclusions. I applaud the efforts of deepening our knowledge of an "orphan" disease, which affects many people and families in the affected areas.

I suggest acceptance for publication.

Response: Thank you for reviewing our article and for your interest in our work.

Reviewer: 3

Reviewer Name: Toshiki Moriyama

Institution and Country: Health and Counseling Center, Osaka University, Japan

Please state any competing interests or state 'None declared': None

Please leave your comments for the authors below

The present manuscript entitled "Prevalence of and risk factors for chronic kidney disease of unknown aetiology in India: secondary data analysis of three population-based cross-sectional studies" by O'Callaghan-Gordo C. et al. reported that rural area, lower education level, higher age, higher HbA1c was associated with prevalence of CKD defined as eGFR <60 mL/min/1.73 m² in the subjects without diabetes, hypertension, and overt proteinuria, using 3 large population-based studies. I think that the findings of large regional differences in prevalence of CKD in India probably provides important information to make a strategy for prevention of CKD. However, this manuscript was so descriptive that it was hard to understand what was the key finding of the present study. I would like to advise the authors to reorganize this manuscript to stress what was the main novel findings. One of the candidates might be effect modification between area and education because it was very difficult to assess a clinical impact of education on the prevalence of CKD in western countries and its impact

was dependent on areas (Table 5). The authors should reorganize their manuscript to make their research question clear.

Response: Thank you for your suggestion.

We have modified the last paragraph of the introduction to make our research question clearer: "(...) the overall aim of the current study was to use a methodology which is comparable to previous studies elsewhere in the world (particularly in Central America) to assess the extent to which reduced kidney function is a problem in India, and which areas and subpopulations are most affected. We therefore: (i) assessed the distribution eGFR and prevalence of eGFR below 60ml/min per 1.73m² (eGFR<60) in Indian populations restricted to those without known risk factors for CKD, i.e. diabetes, hypertension or heavy proteinuria (a marker of primary glomerular disease); ii) compared these outcomes in North and South India and in urban and rural populations; and (iii) identified the risk factors associated with these outcomes."

We have modified the paragraph on results section where the results from the analysis stratified by latitude was presented to clarify results on education." We observed an interaction between the effects of latitude (North/South) and urban/rural residence in association with reduced eGFR (p-value for interaction<0.001). The mean eGFR was lower in rural settings in both Northern and Southern India (controlling for age, sex, education and alcohol intake). However, this decrease was much more marked in Southern India. In Northern India, rural residence, formal education (and duration) and age were the only other risk factor associated with reduced eGFR. In Southern India, being male was also a risk factor for reduced eGFR, whereas formal education was only a risk factor for reduced eGFR among those with more than 10 years of schooling (Table 5). We also observed an interaction between the effects of latitude (North/South) and urban/rural residence in association with eGFR<60 (p-value likelihood-ratio test for interaction<0.001). In Northern India, eGFR<60 was not associated with urban/rural residence, and older age was the only factor associated with eGFR<60. In Southern India, rural residence was the strongest risk factor for eGFR<60 but older age and lower years of formal education also increased the risk of eGFR<60 (Table 5)."

We have also modified the first paragraph of the discussion to stress the main risk factors identified in the current study: "(...) Risk factors of decreased eGFR were different between Southern and Northern India. In Southern India, rural residence, older age and being male were risk factors for both lower mean eGFR and eGFR<60; education was associated with decreased risk for eGFR<60 but not with lower mean eGFR. In Northern India, older age was the only risk factor for both lower mean eGFR and eGFR<60; rural residence and years of formal education were associated with lower mean eGFR but not with eGFR<60. In summary, in Southern India, older age, being male and rural residence were the main risk factors for decreased eGFR, whereas in Northern India older age was the main risk factors for decreased eGFR".

1) Because the majority of the subject was within the normal range of eGFR, I do not think the multivariable-adjusted linear regression models with eGFR as a dependent variable had no clinical value. For example, in Table 2, we should pay our attention not to mean eGFR of education (0, ≤5, 6-10, and >10: 100.7, 105.9, 107.2, and 105.0, respectively), but to the prevalence of eGFR <60 (5%, 1%, 1%, and 1%, respectively). I think that eGFR was not appropriate as an dependent variable in this manuscript, which should be supplementary if the authors should to show.

Response: We agree with the reviewer that eGFR<60 is clinically more relevant than mean eGFR. However, in epidemiological terms, we still consider that it is important to present the results for mean eGFR in the main manuscript (not only as supplementary material). The main aim of the study is to identify risk factors for decreased in kidney function. Models using continuous variables (rather than categorical) have greater statistical power and greater parsimony and therefore are more powerful to identify risk factors than models using categorical variables. Thus, if there is a general 'population shift' in eGFR, this will result in a change in the prevalence of eGFR<60, but will also be reflected in a difference in mean eGFR. Both measures are useful and important epidemiologically, although we agree that eGFR<60 is more relevant clinically. This is a fairly standard approach for continuous

outcome measures of this type (e.g. blood pressure, cholesterol, HbA1c) where a pre-defined 'cut-off' is used clinically, but the continuous measure is also used in epidemiological analyses.

2) To provide the clinical characteristics of patients included in the present study in more details, the main purposes of CARRS study, UDAY study, and ICMR-CHD study.

Response: Study participant's information recorded was slightly different in three studies as each study had different objectives. The objective of CARRS was to study risk factors for cardio metabolic diseases, the objective of UDAY was study risk factors for diabetes and hypertension, and the objective of ICMR-CHD was to study risk factors for cardiovascular disease. BMI, fat free mass, fasting plasma glucose, HbA1c, systolic and diastolic blood pressures, and albumin/creatinine ratio, as well as variables presented in table 2, were the only variables that were available from the three studies. No other clinical variables were recorded in the three studies. In any case, information available was sufficient to conduct the current analysis.

3) Clinical characteristics should be shown after stratified by the key exposure variable, instead of Table 2. How did the authors define rural and urban area and north and south latitude? A map of study area help the readers know the studies places.

Response: We understand that by "clinical characteristics" the reviewer refers to socio-demographic and anthropometric characteristic presented in table 2. There is no 'key exposure variable', although, as stated above, our focus is on: (i) comparing the outcome in North and South India and in urban and rural areas; (ii) investigating risk factors which may explain these population differences. As the main aim of the current analysis is to identify potential risk factors (i.e. key exposures) for decreased levels of eGFR and eGFR<60, we consider that presenting mean levels of eGFR and prevalence of eGFR<60 according to the risk factors under study (socio-demographic and anthropometric characteristic presented in table 2) is relevant. In our opinion, what the reviewer is proposing would be relevant if we already had a key exposure identified, but the purpose of our study is to attempt to identify such key exposure(s).

Study areas were classified in urban or rural according to the definition provided in the 2011 Census on India ("Census of India", 2011). Urban area are defined as: "All places with a municipality, corporation, cantonment board or notified town area committee, etc., and all other places which satisfied the following criteria: A minimum population of 5,000; at least 75 per cent of the male main working population engaged in non-agricultural pursuits; and a density of population of at least 400 persons per km²."

Study areas were classified as North India or South India according to the classification of major geographical areas on India defined by the Indian Council of Medical Research. We have included this information in the revised version of the manuscript: "Finally, we estimated potential interactions between urban (versus rural) residence and latitude (Northern India (i.e. states of Delhi and Haryana) versus Southern India (states of Tamil Nadu and Andhra Pradesh). Classification in concordance with the classification of major geographical areas on India defined by the Indian Council of Medical Research (Longvah et al. 2017)". As suggested, we have included a map in the revised version of the manuscript (figure 1).

4) What was mutual adjustment in Table 4? All covariates should be described in the footnotes of the tables.

Response: Table 4 shows results of the multiple regression analysis. Model 1 included the following variables: area, latitude, education, alcohol consumption, sex and age. Model 2 included exactly the same variables as model 1 (area, latitude, education, alcohol consumption, sex and age), but excluded participants with missing values for fat free mass from the analysis. Model 3 included the same variables plus fat free mass and vegetarianism. We have added this information as a footnote in the revised version of the manuscript.

5) The authors should pay the unit of each covariates included in the multivariable-adjusted model. Coefficients or odds ratio of age (per 1 year) was ridiculous. A risk elevation of a 1-year difference has

no clinical value. Unit of age should be "per 10 year" to estimate the clinically relevant coefficients and odds ratios.

Response: We have taken into consideration the reviewer's comment and we present coefficients or odds ratio of age per 10 years in the revised version of the manuscript. Tables and text have been modified as follows: "As expected, age was an important risk factor for reduced eGFR: eGFR was 9.30 ml/min per 1.73 m² (95%CI=-9.51, -9.09, model adjusted for sex) lower for each additional 10 years of age. Additionally, being male, living in a rural setting, and consuming alcohol were associated with decreased mean eGFR (Table 3). Similarly, the odds of eGFR<60 also increased with age [OR per 10 years, adjusted for sex (95%CI)=2.34 (2.12, 2.59)] and being male, living in a rural setting, living in Southern India and consuming alcohol were also associated with eGFR<60 (Table 3)."

VERSION 2 – REVIEW

REVIEWER	DR HOOI LAI SEONG Nephrologist, Johor Baru, Malaysia
REVIEW RETURNED	26-Aug-2018

GENERAL COMMENTS	Figure 1 should be mentioned in the paper e.g. in line 169. Figure 2 should be in line 175 and Figure 3 in line 199. The tables and the written results tally.
--

REVIEWER	Toshiki Moriyama Health and Counseling Center, Osaka University, Japan
REVIEW RETURNED	12-Sep-2018

GENERAL COMMENTS	The revised manuscript by O'Callaghan-Gordo C. et al. entitled "Prevalence of and risk factors for chronic kidney disease of unknown aetiology in India: secondary data analysis of three population-based cross-sectional studies" reported the prevalence of CKDu in India using 3 large surveys and suggested several clinical contributors to CKDu. As I pointed in my previous review, some findings of the present study provided valuable information to make a health promotion strategy to prevent CKDu in India, but the results were too descriptive. Too many tables with too many rows were very confusing. I advised the authors to pick up several main exposure, such as area, latitude, and education. Were both body mass index and fat free mass necessary necessary in the present study? To assess an association between smoking and the prevalence of CKDu, smoking status should be classified into, at least, 3 categories: never, former, and current smoking. If possible, a dose-dependent association of smoking measure; pack-year before study entry and pack/day at study entry should be assessed. The similar categorization should be essential for an association between alcohol consumption and the prevalence of CKDu. If the main purpose of the present study to identify the potential risk factors of CKDu, these variables should be defined with the clinical values. Because the results of multivariable-adjusted linear regression models and logistic regression models were very similar and eGFR ≥60 mL/min/1.73m² were generally inaccurate, those of linear regression models should be described in supplementary data. Main results should be only those of the associations
--

	between the clinical variables and eGFR < 60 mL/min/1.73m². Tables with too many columns were confusing. "The prevalence ratio of eGFR<60 for rural versus urban residence was higher for participants <50 years than for older groups (Figure 3)." is wrong. Figure 3 had a distribution of prevalence ratio of a single group.
--	--

VERSION 2 – AUTHOR RESPONSE

Reviewer: 1

Reviewer Name: DR HOOI LAI SEONG

Institution and Country: Nephrologist, Johor Baru, Malaysia

Please state any competing interests or state 'None declared': None declared

Please leave your comments for the authors below

1) Figure 1 should be mentioned in the paper e.g. in line 169.

Response: Figure 1 is mentioned in lines 115 and 169

2) Figure 2 should be in line 175 and Figure 3 in line 199.

Response: Figure 2 is in line 175 and Figure 3 is in line 199

3) The tables and the written results tally.

Response: We have carefully reviewed the text and the tables for consistency.

Reviewer: 3

Reviewer Name: Toshiki Moriyama

Institution and Country: Health and Counseling Center, Osaka University, Japan

Please state any competing interests or state 'None declared': None declared.

Please leave your comments for the authors below

The revised manuscript by O'Callaghan-Gordo C. et al. entitled "Prevalence of and risk factors for chronic kidney disease of unknown aetiology in India: secondary data analysis of three population-based cross-sectional studies" reported the prevalence of CKDu in India using 3 large surveys and suggested several clinical contributors to CKDu. As I pointed in my previous review, some findings of the present study provided valuable information to make a health promotion strategy to prevent CKDu in India, but the results were too descriptive. Too many tables with too many rows were very confusing. I advised the authors to pick up several main exposure, such as area, latitude, and education.

1) Were both body mass index and fat free mass necessary in the present study?

Response: Adjustment of the models for fat free mass (and vegetarianism) was important to assess the possibility that differences observed between urban and rural participants were due to differences in diet and/or body composition. We also considered body mass index to provide a better characterization of the study population.

2) To assess an association between smoking and the prevalence of CKDu, smoking status should be classified into, at least, 3 categories: never, former, and current smoking. If possible, a dose-dependent association of smoking measure; pack-year before study entry and pack/day at study entry should be assessed.

Response: The percentage of former smokers was very small (<1%). Therefore, it was not possible to use this category and they were classified as no current smokers. Detailed data on number of cigarettes was not been collected in the surveys.

3) The similar categorization should be essential for an association between alcohol consumption and the prevalence of CKDu. If the main purpose of the present study to identify the potential risk factors of CKDu, these variables should be defined with the clinical values.

Response: Detailed information on quantity of alcohol consumption was not collected in the questionnaires.

4) Because the results of multivariable-adjusted linear regression models and logistic regression models were very similar and eGFR ≥ 60 mL/min/1.73m² were generally inaccurate, those of linear regression models should be described in supplementary data. Main results should be only those of the associations between the clinical variables and eGFR < 60 mL/min/1.73m². Tables with too many columns were confusing.

Response: As commented in the previous revision, we agree with the reviewer that eGFR <60 is clinically more relevant than mean eGFR. However, in epidemiological terms, we still consider that it is important to present the results for mean eGFR in the main manuscript (not only as supplementary material). The main aim of the study is to identify risk factors for decreased in kidney function. Models using continuous variables (rather than categorical) have greater statistical power and greater parsimony and therefore are more powerful to identify risk factors than models using categorical variables. Thus, if there is a general 'population shift' in eGFR, this will result in a change in the prevalence of eGFR <60 , but will also be reflected in a difference in mean eGFR. Both measures are useful and important epidemiologically, although we agree that eGFR <60 is more relevant clinically. This is a fairly standard approach for continuous outcome measures of this type (e.g. blood pressure, cholesterol, HbA1c) where a pre-defined 'cut-off' is used clinically, but the continuous measure is also used in epidemiological analyses.

5) "The prevalence ratio of eGFR <60 for rural versus urban residence was higher for participants <50 years than for older groups (Figure 3)." is wrong. Figure 3 had a distribution of prevalence ratio of a single group.

Response: We have clarified this point in the methods and in the results sections.

Methods section: "We calculated prevalence ratios of eGFR <60 for rural versus urban areas in different age groups".

Results section: "The prevalence ratio of eGFR <60 for rural versus urban residence was higher in participants younger than 50 years (prevalence ratio in age group $\leq 39 = 5.5$, and prevalence ratio in age group 40-49 = 5.8) than in older participants (Figure 3). "

VERSION 3 – REVIEW

REVIEWER	Toshiki MOoriyama Health and Counselling Center, Osaka University, Osaka, Japan
REVIEW RETURNED	29-Nov-2018
GENERAL COMMENTS	Now this manuscript is suitable for publication.